# Witnessing their mother's acute and prolonged stress affects executive functioning in children
Eileen Lashani [1,2,3,4,9] ✉, Isabella G. Larsen[5,9], Philipp Kanske [6,7], Jenny Rosendahl [1], Jost U. Blasberg [1,10] & Veronika Engert [1,2,3,8,10]

Stress can detrimentally affect physical and mental health, especially during childhood. During this critical period, parental bonds can foster resilience or amplify stress. This study explored whether mothers' everyday stress can act as a source of childhood stress, affecting children's executive functioning. 76 healthy mother-child dyads participated, with mothers assigned to a stress-inducing or stress-free condition. Children observed their mothers and were subsequently tested for cognitive flexibility and working memory. Subjective stress, heart rate, and cortisol were measured repeatedly in mothers and children, alongside everyday stress perceptions. Linear mixed models showed that children's acute stress response was associated with impaired cognitive flexibility. Maternal stress, both acute and past-month, was a better predictor of children's cognitive performance than children's own stress. Quadratic relationships indicated the highest error rates at very low and high maternal stress. We found no evidence that children's working memory was impaired by their own or their mothers' stress. Although expected covariations of acute or prolonged stress between mothers and children were not observed, an interaction between maternal past-month stress and acute stress condition provided insights into adaptive mechanisms in children. These findings underscore the significant impact of maternal stress on children's executive functioning, illustrating how parental experiences shape children's everyday outcomes.

Extreme or chronic psychosocial stress can have severe consequences for physical and mental health[1]. In children, stressors are uniquely impactful and can result in detrimental effects on physical, emotional, and cognitive development[2]. During this vulnerable time, parental bonds have the potential to protect against or to exacerbate stress experiences[3,4]. The present study explores this dual role parents can play in shaping their children's stress resilience.

Resilience refers to positive adaptive patterns under adverse circumstances[5] and is alternatingly understood as an a) outcome, b) trait, or c) process[6]. Here, we address resilience as the measurable outcome when facing a stressful situation[6]. Stress can be understood from three

perspectives: a) environmental, focusing on the stressors; b) psychological, focusing on subjective stress perceptions and emotional responses; and c) biological, focusing on the reactivity of physiological systems involved in the stress response[7].

Most research on stress resilience examines outcomes following singular traumatic events[8]. This study, however, hones in on a frequent and highly relevant phenomenon: everyday stress resulting from social interactions. We examined how children's psychophysiological stress response and, as an ultimate outcome, their executive functioning are affected by observing their mothers undergoing an experimental stressor. By investigating how children respond to their mothers' exposure to stress, we aim to

[1]Institute of Psychosocial Medicine, Psychotherapy and Psychooncology, Jena University Hospital, Jena, Germany. [2]German Center for Mental Health (DZPG), partner site Halle-Jena-Magdeburg, Jena, Germany. [3]Center for Intervention and Research on adaptive and maladaptive brain Circuits underlying mental health (C-I-R-C), Jena-Magdeburg-Halle, Jena, Germany. [4]Department of Psychiatry and Psychotherapy, Jena University Hospital, Jena, Germany. [5]Department of Psychology and Neuroscience, Department of Public Policy Studies, Duke University, Durham, NC, USA. [6]Clinical Psychology and Behavioral Neuroscience, Faculty of Psychology, Technische Universität Dresden, Dresden, Germany. [7]Department of Psychology, Faculty of Psychology and Educational Sciences, Babeș-Bolyai University, Cluj-Napoca, Romania. [8]Social Stress and Family Health Research Group, Max-Planck-Institute for Human Cognitive and Brain Sciences, Leipzig, Germany. [9]These authors contributed equally: Eileen Lashani, Isabella G. Larsen.[10]These authors jointly supervised this work: Jost U. Blasberg, Veronika Engert. ✉e-mail: Eileen.Lashani@med.uni-jena.de

understand how caregiver relationships—typically a source of resilience[4,5]—might instead become a risk factor.

Early experiences with attachment figures shape children's ability to interpret social cues and regulate emotions[9–11]. Empathy and physiological synchrony can facilitate this process, enhancing maternal responsivity to the child's needs and strengthening the mother-child bond[12–14]. While an empathic attachment figure ideally enhances a child's stress resilience[9,11], strong empathic tendencies in children may lead to negative outcomes when a child consistently shares their mother's negative emotions or stress.

Drawing from adult studies, we know that stress can be shared empathically between individuals. Such "empathic stress" responses, mirroring the physiological pattern of first-hand stress, include the release of cortisol, increased heart rate, and decreased high frequency heart rate variability (HF-HRV) (see Engert et al. for a review[15]). The extent of empathic stress is influenced by empathic ability as well as physical and emotional closeness[16,17]. Families, with their close bonds and shared living spaces, provide an essential context to investigate empathic stress and its potential maladaptive consequences.

The notion that stress can be transmitted between family members has previously been investigated in the field of family stress research[18,19]. For instance, affect spillover was described from parents' marital conflict or work stress onto their children[20,21]. Beyond self-reported or behavioral indicators of stress transmission, an increasing body of studies has started to assess physiological covariation between mothers and their children in the context of stressful situations[22–24]. Of particular interest are two naturalistic studies; one with six-year-olds, the other with adolescents, which investigated cortisol levels in mother-child dyads in their everyday lives[25,26]. Both revealed significant cortisol covariation in mother-child dyads across the day, particularly during periods of negative affect or stress, and with increased time spent together. Beyond the influence of common stressors, this time-bound covariation suggests the existence of an immediate pathway of stress transmission from one individual to the other.

Accordingly, many studies have examined maternal attunement with their children's stress[24,27,28]. Yet, despite the important implications for childhood outcomes, fewer studies have focused on the reverse—how children physiologically resonate with their stressed mothers. Meaningful examples are two studies with mothers and their 12-14-month-old toddlers, which found that acute stress-induced sympathetic activation in mothers was transferred to their children, particularly when physical touch was involved[29,30].

In a recent study, we extended these findings to 8-12-year-olds, revealing a causal link between psychosocial stress in mothers and children's empathic stress responses[31]. After watching their mothers undergo a standardized psychosocial stress paradigm (Trier Social Stress Test[32]), children were more likely to release cortisol than their peers in a stress-free control group watching their mothers during relaxed reading. They also showed higher subjective stress, state empathy, and HF-HRV responses, with the latter depending on elevated trait cognitive empathy scores[31]. In the current study, the same sample of mother-child dyads was investigated with respect to the children's cognitive performance following observation of their mothers performing the stress or stress-free control tasks.

Since frequent stress can alter the basal activity of the stress system and its responsiveness[2,3], we explored how prolonged stress perceptions in mothers and children might interact with children's acute stress responses. Particularly in childhood, repeated exposure to stressors can lead to adaptations in the stress system, resulting in hypersensitive or attenuated neuroendocrine and psychological responses to acute stressors[33,34].

Like the stress system, cognitive development progresses rapidly during childhood, as children advance through school and face increasing demands to focus, shift attention, and complete complex tasks[35–37]. Early impairment or disrupted development of core executive functions, such as cognitive flexibility and working memory, can have enduring consequences[3,38].

In a meta-analysis, Shields and colleagues[39] found that in adults, acute stress negatively affects cognitive flexibility and working memory, which are

responsible for switching between different tasks or adapting to changing situational requirements[40], and for temporary storage and manipulation of information[41], respectively. Research on the effects of psychosocial stress on these executive functions in children is scarce. Exceptions include a study by Seehagen and colleagues[42], which demonstrated that acute stress diminished cognitive flexibility in children of 15 months of age. Quesada and colleagues[43] found no changes in working memory performance following a psychosocial stressor in 8- to 10-year-old children. The present study investigated both cognitive flexibility and working memory of 8- to 12-year-old children after observing their mothers being exposed to either a laboratory stressor or a stress-free control task.

Studies on chronic stress have shown how early life adversity relates to impaired cognitive flexibility[44–46] and working memory[47–49] stretching into adolescence and adulthood. As mechanisms connecting stress to impaired executive functioning, Marko and Riečansky[50] identified increased sympathetic arousal for impaired cognitive flexibility and cognitive interference for working memory impairment in adults. Additionally, the prefrontal cortex (PFC), a brain region crucial for executive functioning, is highly sensitive to the neuroendocrine environment[51] and has been shown to exhibit disadvantageous structural changes after prolonged psychosocial stress[52].

Of note, the very cognitive functions that are sensitive to stress exposure simultaneously form an integral resource of stress resilience[53]. These findings underscore how impactful childhood stress is for the developing brain and long-term resilient outcomes. However, the mentioned studies on chronic stress focused on circumstances as severe as maltreatment, neglectful caregiving, or poverty. Research on the impact of everyday stressors on children's executive functioning is still needed.

In summary, empathic stress responses among adults have been demonstrated repeatedly[15], initial evidence for covariation between maternal and child stress was found[31], and numerous studies have established a link between stress and impaired executive functioning[39]. We took the next step and sought empirical evidence linking maternal stress, both acute and prolonged, to children's cognitive outcomes. Considering the central role of the mother-child bond for child development and resilient outcomes[4,5,10], alongside the prevalence of chronic stress in families[54,55], understanding how empathic stress influences child-specific outcomes becomes crucial for identifying resilience and risk factors.

Extending Blasberg et al.[31], this study builds on confirmed empathic stress in a German sample of healthy mother-child dyads. We hypothesized that children's acute psychophysiological stress response would impair their cognitive flexibility and working memory. Next to children's stress response, we expected the resonance with their mothers' subjective stress, autonomic reactivity, and cortisol release to predict slower reaction times and reduced accuracy (H1). Second, we explored whether prolonged experiences of everyday stress in children and mothers would influence the effect of the acute experimental condition, differentially affecting children's cognitive performance (H2). As seen for acute stress[31], our third aim was to provide empirical evidence that children's trait empathy (H3a) and relationship closeness (H3b) with their mothers would shape the extent to which they share their mother's past-month stress experience.

## Methods
### Participants

The study sample was recruited within the context of a larger study[31]. Participation was promoted online and with posters, targeting schools and after-school care in Leipzig (Germany) and surrounding areas. Mothers underwent a telephone screening before inclusion. Age criteria were set at 31 to 45 years for mothers and 8 to 12 years for children to minimize hormonal influences on cortisol levels[56]. Mothers after menopause onset and children after menarche onset were excluded. The lower age limit in children ensured they could read and write, show stress responses to a cognitively processed event[57], and show higher-order components of empathy[58]. Further exclusion criteria for mothers were pregnancy, daily cigarette use, regular recreational drug use, or inability to abstain from alcohol for one week. Dyads with a BMI below 18.5 or above 30, lack of German fluency, dyslexia,

**Fig. 1 | Procedure timeline.** The testing session started with a resting period. Children were subsequently familiarized with the cognitive tasks. Simultaneous sampling of all stress markers in mothers and children began 20 minutes before the stress induction (Trier Social Stress Test; TSST; Kirschbaum et al. [32]) or control paradigm. Children then underwent cognitive testing. Finally, mothers and children completed a set of questionnaires. Subjective stress, heart rate, and cortisol were assessed at eight time points each. An electrocardiogram (ECG) was recorded from 20 minutes prior until 45 minutes after the TSST or control task.

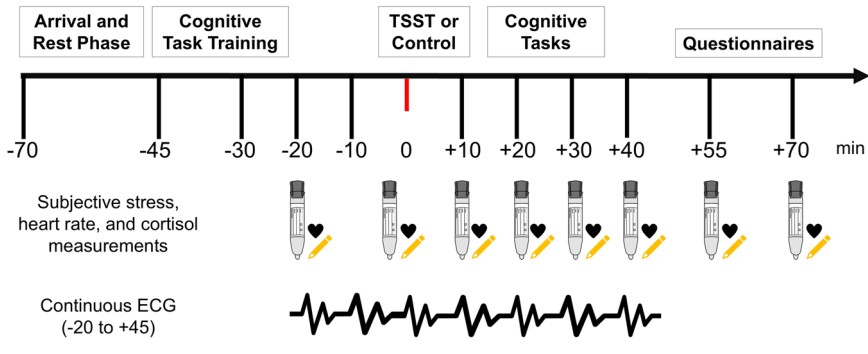

HPA axis-interfering medications, previous psychosocial stress test participation, or other physical and mental health factors that influence the stress response were excluded.

Upon completion of the recruitment phase, 76 children ($n = 37$ girls, $n = 39$ boys) and their mothers were eligible to participate. Children were 8 to 12 years old ($M = 9.95$, $SD = 1.42$), and mothers were 31 to 45 years old ($M = 40.3$, $SD = 3.33$). Dyads were pseudo-randomly assigned to an experimental ($n = 39$) or control group ($n = 37$), controlling for self-reported age and sex. No data on race or ethnicity were collected. The study received approval from the Ethics Committee of Leipzig University (number: 084/18-ek). Written informed consent was sought from mothers on behalf of themselves and their children. Mothers were financially compensated and children received a gift for participation.

## Procedure
The study protocol was registered in the context of the primary study by Blasberg and colleagues[31] (https://osf.io/f3wke). Testing took place at the Max Planck Institute for Human Cognitive and Brain Sciences in Leipzig from June 2018 to March 2020. Testing procedures are depicted in Fig. 1. Each testing session took approximately 2.5 hours and was scheduled between 3 pm and 7 pm to align with cortisol diurnal patterns[59].

Upon arrival, mothers and children were separated, briefed about the experiment, and provided with a snack and juice to equalize blood glucose levels. They were allowed only water for the remainder of the session. After a short resting period to allow participants to settle in, mothers signed informed consent and took a drug test. In parallel, children were familiarized with the cognitive tasks they performed later in the experiment. To reduce potential stress caused by the laboratory setting, time was allotted for resting before beginning the experiment. Children were then accompanied to an observation room which was separated from the testing room by a one-way mirror. Mothers were informed that their children could see and hear the procedure, and instructed for their upcoming task. Depending on the experimental group, mothers either underwent the Trier Social Stress Test (TSST[32]) or a stress-free control task. Upon completion of the mothers' respective tasks, children's cognitive flexibility and working memory were assessed in two computerized tasks, the category-switching[60] and the n-back task[61]. The tasks were administered in randomized order. Before reunion, mothers and children completed a set of questionnaires.

To assess acute stress reactions simultaneously in mothers and children, eight measurements of each subjective stress, heart rate, and salivary cortisol were collected in regular intervals from −20 min until +70 minutes relative to TSST or control task onset. Continuous electrocardiogram (ECG) recordings were collected but not analyzed due to the high number of missing data points[31].

## Stress induction and control tasks
Mothers in the experimental and control groups participated in different tasks of the same duration, providing the children with equivalent exposure to their mothers across groups. In the experimental group, acute psychosocial stress in mothers was induced through the Trier Social Stress Test

(TSST), a standardized laboratory stressor that reliably provokes physiological stress responses[32,62]. The TSST comprises a five-minute anticipation phase (task preparation) and two stressful tasks: a five-minute mock job talk and a five-minute mental arithmetic session. Both are audio- and video-taped and evaluated by a committee of two alleged behavioral analysts, who are trained to act non-responsively.

Mothers in the control group completed a non-stressful reading task, involving a five-minute anticipation phase (silent magazine reading) and a ten-minute task in which they read aloud from a nature book. Unlike the stress group, mothers were alone during the control task. Mothers of both groups were aware that their children were observing.

## Cognitive tasks
To complete the computerized tasks, children sat approximately 50 to 60 cm away from a computer screen with a resolution of 1680 × 1050 pixels and a refresh rate of 60 Hz. Both tasks were implemented using Presentation® (2018).

**Category-switching task.** Cognitive flexibility was assessed using a category-switching task[60]. In three blocks of 64 trials each, children categorized words based on presented categories ("animate/inanimate" or "bigger/smaller than a soccer ball") within 3000 ms. Categories varied randomly, requiring the children to constantly update their mental representation of the task. Accordingly, trial types included repetition and switch trials. Additionally, the Cue Target Interval (CTI) between category and word was either short (200 ms) or long (1000 ms). For an accurate comparison of mean reaction times in switch and repetition trials, the first trial of each block and trials following an error were excluded. Personalized Tukey far out outliers[63] were calculated to exclude reaction times of three standard deviations above or below individual means.

**N-back task.** To assess working memory, the computerized n-back task by Kirchner[61] was used. Children were presented with letters (B, C, D, F, G, J, K, L, M, N, O, P, R, S, T, V, W, Y, Z) for 500 ms in random order. They had to decide whether the current letter equaled the last (1-back) or second last (2-back) letter and had 1500 ms to indicate their decision by pressing the right or left arrow keys. The first block consisted of 65 1-back trials, the second block of 65 2-back trials. Working memory was challenged by holding the series of letters in mind and operationalized in terms of mean reaction time and accuracy per block. Incorrect or missed trials were excluded from the reaction time mean calculations, as were reaction times more than three standard deviations away from the individual mean reaction time (personalized Tukey far outs). As an accuracy index, we calculated d-prime (d') as a normalized comparison of the hit rate (correct responses vs. misses) and the false alarm rate (false rejection vs. correct rejection)[64].

## Measurements of acute stress
**Subjective stress.** Subjective stress was assessed using the single self-report item, "How stressed do you feel at this moment?". Answers were

provided on a 7-point Likert scale ranging from 1 (no stress) to 7 (severe stress).

**Heart rate.** As an indicator of sympathetic activity[65], heart rate was recorded using an OMRON RS2 wrist blood pressure monitor (Medizintechnik Handelsgesellschaft mbH, Mannheim, Germany) that mothers and children were equipped with at the beginning of the testing session. Heart rate, reported in beats per minute (bpm), was checked concurrently with subjective stress and cortisol measurements (see Fig. 1).

**Cortisol.** HPA axis activity was measured in terms of salivary cortisol levels. Saliva was collected using Salivettes (Sarstedt, Nümbrecht, Germany) with a cotton swab that participants held in their mouths for two minutes. The Salivettes were stored at −30 °C and later analyzed at the Biochemical Laboratory of the Department of Biological and Clinical Psychology, Trier University, using a time-resolved fluorescence immunoassay with intra- and inter-assay variabilities of less than 10% and 12%[66].

**Preprocessing.** Heart rate and cortisol data were natural-log-transformed to ensure normal distribution. Cortisol data were winsorized to three standard deviations to adjust for potential outliers[67,68]. To measure stress reactivity, the area under the curve with respect to increase ($AUC_i$[69]) was calculated. The $AUC_i$ has been commonly used in endocrinological research to summarize repeated stress measurements while emphasizing change over time[69]. Following the methodology from Pruessner et al.[69], the $AUC_i$ for each stress marker was calculated by summing the area under the curve of cortisol, subjective stress, and heart rate across the eight measurement time points and subsequently subtracting the area between zero and the baseline measurement.

### Questionnaire data

**Perceived stress in mothers.** As a measure of prolonged stress over the past month, mothers completed the German version of the "Perceived Stress Scale" (PSS[70]). It comprises ten items, assessing unpredictability, uncontrollability, and overload in their lives over the past month. Items are rated on a Likert-type scale ranging from 0 ("never") to 4 ("very often"). Total scores therefore range between 0-40 with higher scores indicating higher stress levels. In our sample, internal consistency was acceptable (α = 0.76).

**Perceived stress in children.** Children completed the German version of the "Perceived Stress Scale for Children" (PSS-C[71]), featuring 13 age-adapted items assessing prolonged stress. To yield more adequate reports in that age group, the questionnaire refers to the past week rather than month. The answer scale ranges from 0 ("never") to 3 ("a lot") and is complemented by a graphic representation of quantities. Total scores range between 0-39 with higher scores indicating higher stress levels. The questionnaire showed poor internal consistency in our sample (α = 0.5). This issue likely arose because the questionnaire taps into different areas of children's lives, such as school performance, friendships, relationships with parents, and emotional states. As the goal was to gauge accumulated everyday stress, using a sum score of these heterogeneous items was preferred over discarding the questionnaire at whole. However, the potential lack of reliability and validity calls for cautious interpretation of PSS-C results.

**Trait empathy.** As a potential modulator of stress transmission between mothers and children, we evaluated children's trait empathy using the "Empathy Questionnaire for Children and Adolescents" (EmQue-CA[72]). Children answered the 14 items on a three-point scale (1: "not true" to 3: "true"). The EmQue-CA provides an overall trait empathy score (α = 0.67) and the three subscales: affective empathy (α = 0.54), cognitive empathy (α = 0.76), and intention to comfort (α = 0.47). Higher scores

reflect higher trait empathy on the respective scale. As only cognitive empathy showed acceptable reliability, this subscale alone was considered when testing moderation hypothesis H3a.

**Relationship closeness.** As an indicator of self-reported closeness between mothers and children, all participants filled out the Inclusion of Other in the Self Scale (IOS[73]). This single-item graphic scale contains seven Venn diagram-like pairs of circles with varying overlap. Mothers and children selected the circle pair that best represented their relationship with each other. Scores could lie between 1-7 with higher values indicating more relationship closeness.

### Statistical analysis

Statistical analyses were conducted using R 4.2.0[74]. The analyses for this study were not preregistered. All analysis scripts and data required to replicate the results are publicly available at https://osf.io/kyrbh/. The sample size was predetermined based on the power analysis reported in Blasberg et al.[31], aiming to detect medium to large effects of empathic stress[17]. Given the exploratory nature of the study, we deemed the sample size adequate for further analyses of empathic stress effects on cognition, acknowledging that power might be limited for some models. Statistical significance was tested two-sided with an alpha of 0.05 in all analyses. Assumptions for each type of analysis, including normality and homoscedasticity, were tested using visual inspection and considered adequately met unless specified otherwise.

Prior to model testing, children with insufficient task understanding were excluded, resulting in the exclusion of one child for the category switching task and two children for the n-back task. Further, individuals with two or more consecutive missing data points of a stress marker were excluded for the respective analysis. For cases with only one missing value, the average of the two data points surrounding the missing value was imputed. Subjective stress data from two individuals ($n = 1$ mother and $n = 1$ child) and heart rate data from one child were missing from one time point. Five individuals missed cortisol data from one time point ($n = 2$ mothers and $n = 3$ children), with one mother missing cortisol data from two consecutive time points.

**Group differences.** Based on our previously published results in Blasberg et al.[31] we knew that the number of mothers and children exhibiting a physiologically significant cortisol release ( > 1.5 nmol/l[75]) was higher in the stress group (28 mothers; seven children) than in the control group (six mothers; one child). Also, stress group mothers and children reported higher levels of subjective stress, and stress group mothers exhibited higher heart rate reactivity than their control group counterparts. Our stress induction paradigm was therefore successful.

In addition to this manipulation check, we tested whether performance on the cognitive tasks differed between stress and control groups. Repeated-measures ANOVAs were conducted for accuracy and reaction time, considering multiple values per participant due to the repeated measures task structures. Accuracy was conceptualized as error rates for the category-switching task[60] and as a d' index for the n-back task[61]. Reaction time was assessed in milliseconds in both tasks. Two additional ANOVAs were run to test whether the stress manipulation influenced mothers' and children's self-report of recently perceived stress.

**Linear mixed models.** Linear mixed models (LMMs) were employed to examine relationships between children's and mothers' stress measures and cognitive performance. Two model-versions were run per hypothesis and task, using either reaction time or accuracy as dependent variables. Fixed effects included children's age, sex, task characteristics (trial or block types), and experimental group in all models. For each hypothesis, we further added fixed effects of interest. All numeric predictors were scaled prior to analysis to enhance interpretability. Each model incorporated a random intercept for subjects to account for the repeated measures structure of the cognitive tasks.

By adding fixed effects of children's stress reactivity, measured by the $AUC_i$ for a given marker, the influence of overall child stress on their cognitive performance was indicated. To target the effect of children's acute resonance with their mother's stress on cognitive performance (H1), an interaction term of child and maternal $AUC_i$s of either cortisol, subjective stress, or heart rate was introduced. This resulted in three models per outcome and task. The hypothesized modulating effect of recent everyday stress (H2) was tested by adding a three-way interaction of children's perceived stress (PSS-C), mothers' perceived stress (PSS), and experimental group.

As outlined above, high levels of stress can have adverse effects on cognitive performance. However, there is reason to assume that the full spectrum of stress, including low and moderate levels, can relate to cognitive performance in a linear or in a quadratic fashion[76]. Therefore, we first conducted the analysis using the linear term for a given child or maternal stress marker, adding quadratic terms to each model in a second exploratory step.

All models were fitted using the "lme4" R package[77] and *p*-values were computed using the "lmerTest" package[78]. Bootstrap procedures with 1000 resamples were applied for robust parameter estimation ("boot" package[79,80]). Continuous predictors were z-standardized to address multicollinearity, yielding acceptable variance inflation factors (VIF) of < 5 in all models. Given the number of models, careful considerations were made regarding the correction for multiple comparisons. Following recommendations by García-Pérez[81], we concluded that our analyses do not call for correction, as we assume no omnibus null hypothesis. Instead, we explore the effects of different stress systems (i.e., subjective psychological stress, autonomically regulated heart rate, and HPA-axis output cortisol) on different outcomes (i.e., reaction time and accuracy), which can be assumed to demonstrate opposing patterns in terms of a speed-accuracy trade-off[82].

**Moderation analysis.** This analysis tested whether the hypothesized relationship between maternal and child prolonged stress was moderated by children's empathy and closeness to their mothers. Multiple linear regression models were used. In the basic model, maternal prolonged stress (PSS) predicted children's prolonged stress (PSS-C), controlling for age and sex. For the moderation hypothesis H3a, an interaction term of PSS and children's cognitive empathy was introduced. For H3b the interaction of PSS and relationship closeness (IOS) was added to the basic model.

### Reporting summary
Further information on research design is available in the Nature Portfolio Reporting Summary linked to this article.

## Results
Correlations between all predictor and outcome variables are provided in Supplementary Table 1. For more detail on sample characteristics and group differences in stress responses, please see Blasberg et al.[31].

### Group differences in cognitive performance and prolonged stress
The repeated-measures ANOVA detected no significant effects of experimental condition (TSST vs. control) on children's reaction time or accuracy in the category-switching ($F_{(1,73)} = 0.28$, $p = 0.599$, $d = -0.1$, 95% CI [−0.33, 0.12]; $F_{(1,73)} = 1.37$, $p = 0.245$, $d = 0.24$, 95% CI [0.01, 0.47]) or n-back tasks ($F_{(1,72)} = 0.03$, $p = 0.853$, $d = -0.04$, 95% CI [−0.36, 0.28]; $F_{(1,70)} = 0.33$, $p = 0.565$, $d = 0.08$, 95% CI [−0.25, 0.4]). Thus, we did not find evidence that the stress manipulation affected children's executive functioning on a group level.

ANOVA results revealed no significant difference in maternal and child prolonged stress between the TSST and control groups (PSS: $F_{(1,72)} = 0.04$, $p = 0.851$, $d = -0.04$, 95% CI [−0.5, 0.41]; PSS-C: $F_{(1,70)} = 0.72$, $p = 0.400$, $d = -0.2$, 95% CI [−0.66, 0.26]). Therefore, we have no indication to believe that the stress manipulation compromised the

reliability of stress reports. The results of the ANOVA models are summarized in Supplementary Table 2.

### Effects of acute stress on cognitive performance
**Reaction time.** For the category-switching task, analyses showed no significant effects of children's and mothers' stress markers, or of their interaction, on reaction time. Test statistics of all models predicting reaction time in the category-switching task are provided in Supplementary Table 3. In contrast, reaction time in the n-back task was significantly predicted by maternal subjective stress in the linear model ($t_{(67)} = -2.41$, $p = 0.017$, $\beta = -53.64$, 95% CI [−100.15, −11.48]). This effect suggests increased response speed when mothers reported more subjective stress. No additional linear or quadratic stress markers of either children or mothers significantly predicted reaction time in the working memory task as outlined in Supplementary Table 4.

**Accuracy.** Regarding accuracy, children's error rates in the category-switching task were related to mothers' acute stress reactivity, showing more errors in children with stronger maternal cortisol increase (Table 1a). We found no evidence of an interaction between mother and child cortisol reactivity and therefore of an influence of empathic stress resonance on children's cognitive flexibility. No other markers, including those of children, showed significant linear relationships with children's performance.

Introducing quadratic terms substantially changed this pattern (Table 1b). Children's subjective stress predicted optimal accuracy in the cognitive flexibility task around average stress reactivity (Fig. 2a). While children's heart rates remained an insignificant predictor (Fig. 2b), an effect of their cortisol reactivity on error rates emerged (Fig. 2c). All quadratic maternal stress markers significantly predicted children's cognitive flexibility, with optimal accuracy ranging between 1 SD and 0.5 SD below average stress reactivity (Fig. 2d–f). As before, we did not find the hypothesized interaction between child and maternal stress markers, leaving us with a lack of evidence for empathic stress resonance effects on cognitive flexibility. Moreover, no effects of child stress, maternal stress, or their interaction were found on accuracy in the n-back task (see Supplementary Table 5).

**Summary acute stress.** Taken together, we found evidence that children's acute stress reactivity, particularly subjective stress and cortisol release, was associated with impaired accuracy but not reaction time in the category-switching task. Notably, all maternal stress markers predicted children's performance on the same outcome, indicating impairing effects of very low or very high maternal stress. Further, maternal subjective stress related to enhanced reaction time in the n-back task. While the data did not support the hypothesized empathic stress resonance effects on cognition, maternal acute stress emerged as a key predictor of children's performance.

### Effects of prolonged stress on cognitive performance
**Reaction time.** The fixed effects predicting cognitive flexibility speed are presented in Table 2 and reaction time results for working memory are detailed in Supplementary Table 6. As with acute stress, linear and quadratic analyses did not reveal significant associations between maternal or child prolonged stress and reaction time in the category-switching task. Further, no associations were found with speed in the working memory task.

**Accuracy.** Results for accuracy in the category-switching task are presented in Table 2. We found that children's error rate was predicted by maternal past-month stress, which exhibited a quadratic effect with optimal performance around 0.25 SD above average (Fig. 3a). Further, an interaction effect of condition with maternal prolonged stress emerged (Fig. 3b). It shows that lower levels of maternal stress in the past month related to better accuracy in the control condition but worse accuracy in

**Table 1 | Linear mixed model results for (a) linear and (b) quadratic effects of acute stress on children's error rate**

| | Subjective stress | | | | | | Heart rate | | | | | | Cortisol | | | | | |
|---|---|---|---|---|---|---|---|---|---|---|---|---|---|---|---|---|---|---|
| | β | SE | $CI_{lower}$ | $CI_{upper}$ | t-value | p-value | β | SE | $CI_{lower}$ | $CI_{upper}$ | t-value | p-value | β | SE | $CI_{lower}$ | $CI_{upper}$ | t-value | p-value |
| **(a)** | | | | | | | | | | | | | | | | | | |
| Intercept | 0.13 | 0.02 | 0.08 | 0.17 | 5.66 | **< 0.001** | 0.13 | 0.02 | 0.08 | 0.17 | 5.9 | **< 0.001** | 0.15 | 0.02 | 0.11 | 0.19 | 6.96 | **< 0.001** |
| Age | −0.06 | 0.01 | −0.08 | −0.03 | −4.81 | **< 0.001** | −0.06 | 0.01 | −0.08 | −0.04 | −4.79 | **< 0.001** | −0.06 | 0.01 | −0.09 | −0.04 | −5.74 | **< 0.001** |
| Sex | −0.05 | 0.02 | −0.1 | 0 | −2.04 | **0.042** | −0.05 | 0.02 | −0.1 | −0.01 | −2.21 | **0.028** | −0.06 | 0.02 | −0.1 | −0.02 | −2.84 | **0.005** |
| CTI | 0.02 | 0.01 | 0.01 | 0.04 | 3.32 | **0.001** | 0.02 | 0.01 | 0.01 | 0.04 | 3.32 | **0.001** | 0.02 | 0.01 | 0.01 | 0.04 | 3.32 | **0.001** |
| Switch | 0.07 | 0.01 | 0.06 | 0.09 | 10.68 | **< 0.001** | 0.07 | 0.01 | 0.06 | 0.09 | 10.68 | **< 0.001** | 0.07 | 0.01 | 0.06 | 0.09 | 10.68 | **< 0.001** |
| Condition | 0.02 | 0.03 | −0.03 | 0.08 | 0.81 | 0.418 | 0.03 | 0.03 | −0.02 | 0.08 | 0.98 | 0.327 | −0.01 | 0.03 | −0.06 | 0.04 | −0.35 | 0.73 |
| Child $AUC_i$ | 0 | 0.01 | −0.03 | 0.03 | −0.26 | 0.798 | −0.01 | 0.01 | −0.04 | 0.01 | −1.15 | 0.25 | 0.02 | 0.01 | 0 | 0.04 | 1.81 | 0.071 |
| Mother $AUC_i$ | 0.01 | 0.01 | −0.02 | 0.04 | 0.91 | 0.365 | 0.01 | 0.01 | −0.01 | 0.04 | 1.07 | 0.287 | 0.04 | 0.01 | 0.02 | 0.07 | 3.35 | **0.001** |
| Child*mother $AUC_i$ | −0.01 | 0.01 | −0.04 | 0.01 | −1.25 | 0.213 | 0 | 0.01 | −0.02 | 0.01 | −0.46 | 0.648 | 0 | 0.01 | −0.02 | 0.02 | 0.11 | 0.914 |
| **(b)** | | | | | | | | | | | | | | | | | | |
| Intercept | 0.11 | 0.02 | 0.07 | 0.14 | 5.46 | **< 0.001** | 0.1 | 0.02 | 0.05 | 0.15 | 4.13 | **< 0.001** | 0.11 | 0.03 | 0.06 | 0.17 | 4.32 | **< 0.001** |
| Age | −0.05 | 0.01 | −0.07 | −0.03 | −5.04 | **< 0.001** | −0.06 | 0.01 | −0.09 | −0.04 | −4.87 | **< 0.001** | −0.06 | 0.01 | −0.08 | −0.04 | −5.87 | **< 0.001** |
| Sex | −0.06 | 0.02 | −0.1 | −0.02 | −2.79 | **0.006** | −0.05 | 0.02 | −0.09 | 0 | −1.97 | **0.05** | −0.05 | 0.02 | −0.1 | −0.01 | −2.44 | **0.015** |
| CTI | 0.02 | 0.01 | 0.01 | 0.04 | 3.32 | **0.001** | 0.02 | 0.01 | 0.01 | 0.04 | 3.32 | **0.001** | 0.02 | 0.01 | 0.01 | 0.04 | 3.32 | **0.001** |
| Switch | 0.07 | 0.01 | 0.06 | 0.09 | 10.68 | **< 0.001** | 0.07 | 0.01 | 0.06 | 0.09 | 10.68 | **< 0.001** | 0.07 | 0.01 | 0.06 | 0.09 | 10.68 | **< 0.001** |
| Condition | −0.01 | 0.03 | −0.06 | 0.04 | −0.24 | 0.808 | 0.03 | 0.03 | −0.02 | 0.08 | 1.04 | 0.3 | 0.01 | 0.03 | −0.04 | 0.06 | 0.33 | 0.74 |
| Child $AUC_i$ | 0 | 0.01 | −0.03 | 0.02 | −0.4 | 0.691 | −0.02 | 0.01 | −0.04 | 0.01 | −1.3 | 0.194 | 0.03 | 0.01 | 0 | 0.05 | 2.25 | **0.025** |
| Mother $AUC_i$ | 0.03 | 0.01 | 0 | 0.05 | 1.78 | 0.076 | 0.01 | 0.01 | −0.01 | 0.04 | 0.89 | 0.376 | 0.02 | 0.02 | −0.01 | 0.05 | 1.47 | 0.143 |
| Child $AUC_i^2$ | 0.02 | 0.01 | 0.01 | 0.03 | 3.58 | **< 0.001** | 0 | 0.01 | −0.01 | 0.01 | 0.24 | 0.812 | 0 | 0.01 | −0.02 | 0.02 | −0.08 | 0.936 |
| Mother $AUC_i^2$ | 0.02 | 0.01 | 0.01 | 0.03 | 3.31 | **0.001** | 0.02 | 0.01 | 0 | 0.04 | 2.13 | **0.034** | 0.02 | 0.01 | 0.01 | 0.04 | 2.55 | **0.011** |
| Child*mother $AUC_i$ | 0.01 | 0.01 | −0.01 | 0.03 | 0.62 | 0.537 | −0.01 | 0.01 | −0.03 | 0.01 | −0.96 | 0.337 | −0.01 | 0.01 | −0.03 | 0.02 | −0.6 | 0.548 |

All numeric predictors were scaled prior to analysis. The β-values represent the standardized fixed effects predicting children's error rate. The β-values represent the standardized fixed effects predicting children's error rate in the category-switching task (Mayr & Kliegl, [66]). Fixed effects of interest include the AUGs (Area Under the Curve with respect to increase) of children's and mothers' stress markers. Standard errors (SE), 95% confidence intervals (CI), t-values, and p-values are reported. All values were obtained using bootstrapping with 1000 iterations. Random intercepts were specified for participants. Results reaching the significance threshold of $p < 0.05$ were marked in bold.

**Fig. 2 | Linear and quadratic effects of acute stress markers on children's error rates.** Effects of the AUC$_i$s (Area Under the Curve with respect to increase) of children's and mothers' stress markers on children's error rates in the category-switching task (Mayr & Kliegl, [60]) are depicted. Panels **a–c** represent the linear and quadratic effects of children's subjective stress, heart rate, and cortisol. Panels **d–f** represent the corresponding effects of maternal stress markers. The grey shading represents the standard error (*SE*). Four data points per dyad ($n = 75$) are shown due to the 2×2 task structure.

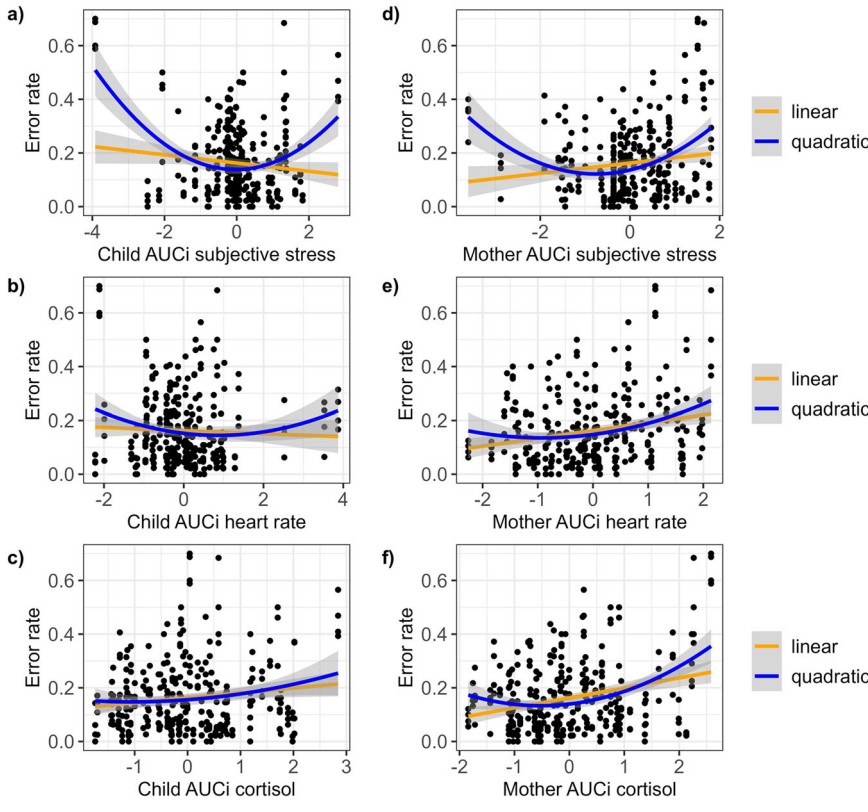

the TSST condition. This pattern reversed with higher maternal prolonged stress. Here, children's cognitive flexibility was better in the TSST condition than in the control condition. Paralleling results for acute stress, we found no evidence for effects of children's or mothers' prolonged stress on accuracy in the n-back task (Supplementary Table 6).

**Summary prolonged stress.** In summary, the results demonstrate that maternal prolonged stress was related to children's accuracy in the category-switching task. No evidence was found for effects on children's reaction time in the category-switching task or on any of the outcomes in the n-back task. Further, the analyses indicate that mothers' latent stress levels modified the impact of an acute stressor on their children's cognitive flexibility. While the hypothesized interaction with children's stress levels (and thus resonance) was not observed, these results resemble the pattern found for acute stress.

### Relationship of mother and child stress
Neither the model featuring children's empathy, nor the model using dyad closeness explained a significant amount of variance in children's everyday stress perceptions, accounting for only 8.5% and 6% of variance, respectively ($F(5, 68) = 1.26$, $p = 0.29$, $R^2 = 0.085$, 95% CI [0.043, 0.283]; $F(5, 68) = 0.86$, $p = 0.511$, $R^2 = 0.06$, 95% CI [0.03, 0.348]). Since no evidence for a relationship between maternal and child prolonged stress was found, H3 was rejected.

### Discussion
This study investigated how witnessing their mothers under stress can pose a challenge to children's own stress resilience by influencing their executive functioning. To this end, children observed their mothers completing either a psychosocial stress or a stress-free control task and subsequently underwent cognitive testing. Subjective stress, heart rate, and cortisol of both mothers and children were collected throughout the session, as well as perceived stress in the past month to gauge prolonged stress load. We analyzed the relationship between these stress markers and children's

cognitive performance, and between children's and mothers' prolonged stress. While former studies have shown that children resonate emotionally and physiologically with their stressed mothers[29–31], and that both acute[42] and chronic[3,38] stress exposure affect children's cognition, an explicit link between maternal stress exposure and children's cognitive performance requires empirical evidence. Here we addressed this gap, contributing to the understanding of how the mother-child bond can represent not only a source of - but also a strain on - children's stress resilience.

Our results demonstrate that children's acute stress markers were only sporadically associated with impaired cognitive performance and no effects were found for their self-reported prolonged stress. Interestingly, it was maternal acute and prolonged stress that emerged as the better predictor of children's executive functioning, especially when considering a quadratic relationship. While we found no evidence for the hypothesized effects of mother-child resonance on children's cognitive performance, an interaction between maternal past-month stress and acute experimental condition pointed towards adaptive mechanisms in children with frequently stressed mothers. Overall, our results show that children's executive functions are sensitive to their mothers' stress experiences.

### Acute stress effects on cognitive performance
The present study extends our previous finding of a successful empathic stress induction in children[31] by exploring the effects of this second-hand response on executive functioning. Although children observing their mothers undergo the TSST experienced subjective and physiological stress more frequently compared to the control group[31], this did not translate into a statistical group difference in cognitive performance. On an individual level, however, we found several stress effects of both children and mothers on executive functioning. Of note, these effects followed a general pattern, such that stress effects on cognitive flexibility were only detectable for accuracy, whereas stress effects on working memory were only found for reaction time. This observation is in line with previous studies[83,84].

In the category-switching task, children's cortisol response was linked to impaired accuracy, aligning with a body of studies underscoring the role

**Table 2 | Linear mixed model results for (a) linear and (b) quadratic effects of prolonged stress on children's reaction time and error rate**

| a) | Reaction time | | | | | | Error rate | | | | | |
|---|---|---|---|---|---|---|---|---|---|---|---|---|
| | β | *SE* | CI$_{lower}$ | CI$_{upper}$ | *t*-value | *p*-value | β | *SE* | CI$_{lower}$ | CI$_{upper}$ | *t*-value | *p*-value |
| Intercept | 1023.3 | 41.52 | 941.31 | 1100.19 | 24.65 | **<0.001** | 0.13 | 0.02 | 0.08 | 0.16 | 6.17 | **<0.001** |
| Age | −46.6 | 25.39 | −93.76 | 5.22 | −1.84 | 0.068 | −0.05 | 0.01 | −0.08 | −0.03 | −4.35 | **<0.001** |
| Sex | 139.34 | 48.28 | 43.32 | 233.71 | 2.89 | **0.004** | −0.05 | 0.02 | −0.1 | 0 | −2.1 | **0.037** |
| CTI | 128.83 | 12.78 | 103.63 | 153.41 | 10.08 | **<0.001** | 0.02 | 0.01 | 0.01 | 0.04 | 3.39 | **0.001** |
| Switch | 165.44 | 12.75 | 140.43 | 190.54 | 12.97 | **<0.001** | 0.07 | 0.01 | 0.06 | 0.09 | 10.7 | **<0.001** |
| Condition | −32.85 | 50.33 | −126 | 67.71 | −0.65 | 0.514 | 0.03 | 0.02 | −0.01 | 0.08 | 1.25 | 0.211 |
| PSS-C | −25.89 | 35.03 | −90.67 | 44.24 | −0.74 | 0.461 | 0 | 0.02 | −0.03 | 0.04 | 0.12 | 0.903 |
| PSS | 29.36 | 34.61 | −38.81 | 94.63 | 0.85 | 0.397 | 0.01 | 0.02 | −0.02 | 0.05 | 0.83 | 0.409 |
| Condition*PSS-C | −38.98 | 48.03 | −129.16 | 52.61 | −0.81 | 0.418 | 0.01 | 0.02 | −0.04 | 0.05 | 0.32 | 0.749 |
| Condition*PSS | −70.44 | 50.66 | −166.75 | 27.67 | −1.39 | 0.166 | −0.05 | 0.02 | −0.1 | 0 | −2 | **0.046** |
| PSS-C*PSS | 21.57 | 45.89 | −64.01 | 113.86 | 0.47 | 0.639 | −0.02 | 0.02 | −0.06 | 0.02 | −1.05 | 0.293 |
| Condition*PSS-C*PSS | 4.73 | 56.75 | −100.66 | 120.58 | 0.08 | 0.934 | 0.02 | 0.03 | −0.03 | 0.08 | 0.91 | 0.366 |
| **b)** | | | | | | | | | | | | |
| Intercept | 1062.61 | 47.62 | 970.1 | 1150.1 | 22.31 | **<0.001** | 0.1 | 0.02 | 0.06 | 0.14 | 4.53 | **<0.001** |
| Age | −55.82 | 26.05 | −106.01 | −4.31 | −2.14 | **0.033** | −0.05 | 0.01 | −0.07 | −0.03 | −4.05 | **<0.001** |
| Sex | 138.55 | 50.07 | 41.2 | 234.56 | 2.77 | **0.006** | −0.06 | 0.02 | −0.11 | −0.02 | −2.67 | **0.008** |
| CTI | 128.83 | 12.78 | 103.63 | 153.41 | 10.08 | **<0.001** | 0.02 | 0.01 | 0.01 | 0.04 | 3.39 | **0.001** |
| Switch | 165.44 | 12.75 | 140.43 | 190.54 | 12.97 | **<0.001** | 0.07 | 0.01 | 0.06 | 0.09 | 10.7 | **<0.001** |
| Condition | −26 | 50.17 | −124.17 | 72.37 | −0.52 | 0.605 | 0.03 | 0.02 | −0.01 | 0.08 | 1.31 | 0.19 |
| PSS-C | −1.83 | 39.57 | −77.63 | 79.72 | −0.05 | 0.963 | 0.01 | 0.02 | −0.03 | 0.04 | 0.3 | 0.766 |
| PSS | 25.8 | 34.63 | −43.36 | 92.91 | 0.75 | 0.457 | 0.02 | 0.02 | −0.01 | 0.05 | 1.08 | 0.281 |
| PSS-C² | −29.65 | 19.03 | −65.84 | 7.97 | −1.56 | 0.12 | 0 | 0.01 | −0.01 | 0.02 | 0.18 | 0.859 |
| PSS² | −14.86 | 20.92 | −55.46 | 26.57 | −0.71 | 0.478 | 0.03 | 0.01 | 0.01 | 0.05 | 2.74 | **0.007** |
| Condition*PSS-C | −49.32 | 51.01 | −148.78 | 43.01 | −0.97 | 0.334 | −0.01 | 0.02 | −0.05 | 0.04 | −0.26 | 0.795 |
| Condition*PSS | −62.47 | 51.44 | −158.7 | 39.29 | −1.21 | 0.226 | −0.06 | 0.02 | −0.1 | −0.01 | −2.5 | **0.013** |
| PSS-C*PSS | 39.01 | 48.98 | −55.22 | 133.11 | 0.8 | 0.426 | −0.02 | 0.02 | −0.06 | 0.03 | −0.78 | 0.438 |
| Condition*PSS-C*PSS | 3.35 | 56.45 | −99.33 | 117.85 | 0.06 | 0.953 | 0.03 | 0.03 | −0.02 | 0.08 | 0.96 | 0.337 |

All numeric predictors were scaled prior to analysis. The β-values represent the standardized fixed effects predicting reaction time and error rate in the category-switching task. Fixed effects of interest include the PSS (Perceived Stress Scale; Cohen et al. [70]) and the PSS-C (Perceived Stress Scale for Children; White, [71]) as markers of children's and mothers' prolonged stress. Standard errors (*SE*). 95% confidence intervals (*CI*), *t*-values, and *p*-values are reported. All values were obtained using bootstrapping with 1000 iterations. Random intercepts were specified for participants. Results reaching the significance threshold of $p < 0.05$ were marked in bold.

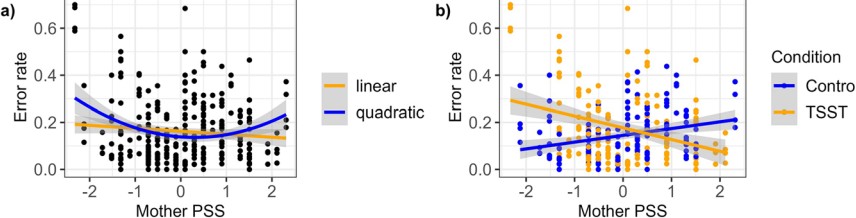

**Fig. 3 | Main and interaction effects of maternal prolonged stress on children's error rates.** In **a**, main effects of mothers' PSS (Perceived Stress Scale; Cohen et al. [70]) scores on children's error rates in the category-switching task (Mayr & Kliegl, [60]) are depicted. In **b**, interaction effects of the PSS with the experimental condition, i.e. a stress-free control task vs. the TSST (Trier Social Stress Test, Kirschbaum et al. [32]), on children's error rates are shown. The grey shading represents the standard error (*SE*). Four data points per dyad (*n* = 75) are shown due to the 2×2 task structure.

of cortisol in detrimental stress effects on executive functioning[51,83]. Through the exploratory second step of the data analysis, accounting for a possible quadratic relationship, children's subjective stress reaction emerged as an additional predictive marker of cognitive flexibility, revealing that a mild stress response was linked to improved accuracy. The notion that stress and cognitive performance can relate in a bell-shaped manner has long been discussed in the context of the Yerkes-Dodson law[85]. Despite the intuitive appeal of the idea that an ideal amount of stress increases performance, this assumption has been criticized for its lack of empirical evidence[86]. Yet, there is reason to assume that acute stress can be beneficial, as one of the core functions of the human stress response is to mobilize energy[2]. On a cognitive

level, mild stress has also been shown to relate to optimal PFC functioning[51] and to narrow down the attentional focus to the task at hand[87].

Interestingly, we found similar quadratic associations for all maternal stress markers and a linear impairing effect of maternal cortisol on the same outcome. These effects occurred regardless of children's stress reactions. In our experiment, children's stress responses were presumably provoked by their empathic reaction to their mothers' stress. We showed this to be the case in the original report by Blasberg and colleagues[31], where covariance between maternal and child stress depending on children's empathy was found. While the absence of a statistically significant interaction effect between maternal and child stress in our study challenges the assumption that acute empathic stress resonance is the cause of cognitive flexibility impairment, maternal subjective and physiological stress emerged as key predictors of children's performance.

This partially aligns with Roos et al.'s[88] discovery that self-reported maternal parenting stress was associated with children's physiological stress reactivity, which, in turn, was *not* associated with children's cognitive impairment. Parenting behavior has frequently been discussed as an explanatory pathway by which parental stress exerts effects on children[89]. If a parent can provide high levels of protection[90], positive bonding experiences[91], and secure attachment[92], their parenting behavior can act as a protective mechanism, fostering resilience in the face of stress and the development of executive functioning. However, as our findings illustrate, if the child is subjected to their parents experiencing stressful situations, the child is at risk of suffering the negative second-hand effects, even if not acutely resonating with parental stress.

Whereas mothers' and children's stress markers predominantly showed the hypothesized impairing relationship with cognitive flexibility, the effects of acute stress on working memory demonstrated contradicting patterns. Children's response speed increased with higher maternal subjective stress, while no evidence for effects of their own stress responses was found. The current understanding of stress effects on working memory speed remains inconclusive, with studies reporting both positive[93] and negative[84] effects, particularly lacking research in children except for one null finding[43]. Although energy mobilization and attention narrowing may explain speed enhancement effects of stress, the present data don't allow for a conclusive interpretation regarding this matter.

In summarizing our findings on acute stress effects on executive functioning, the key question arises: How does maternal subjective and physiological stress impact child cognition if the child does not acutely resonate with the mother?

## Prolonged stress effects on cognitive performance

A likely mechanism linking maternal stress and children's cognition lies in the influence of general stress reactivity patterns. Research investigating the influence of subjective appraisals of stressful situations on the extent of the acute physiological stress response has shown that individuals perceiving stressors as more threatening or overwhelming in general, show increased psychophysiological responses to acute stressors[94,95]. Thus, if mothers who show stronger stress responses in the TSST are also more stress-reactive in everyday life[96], it is plausible that their children are more frequently exposed to a stressed parent and experience more stress overall. However, children's self-reported prolonged stress was not significantly related to their cognitive performance, contradicting the consensus that prolonged stress negatively affects cognitive functioning[1,2]. Since this association can be considered established, we have reason to believe that the questionable internal consistency of the PSS-C[71] may have compromised the validity of our prolonged stress measure for children. Additionally, none of the children reported high levels of perceived stress on this scale. For these reasons, we cannot conclusively interpret the lack of evidence for an interaction between maternal and children's prolonged stress.

Instead of the expected interaction effect of mother and child prolonged stress, we found that maternal perceived stress in the past month predicted performance in the category-switching task irrespective of children's PSS-C values. As an influential factor of this relationship, Knauft and

colleagues[97] highlighted the importance of the interplay between chronic and acute stress. In their study with adults, they found that an acute stressor had negative effects on cognitive flexibility only in the absence of chronic stress. Our results affirm this pattern: Children's cognitive flexibility in the stress group was only impaired when their mothers did not report prolonged stress. This pattern may indicate the presence of successful coping, with children in high-stress environments being less perturbed by their parent's acute stress, having adapted to and regulated or repressed their emotional reactions. Especially in younger children, whose internal working models are still developing, emotional detachment from their caregivers can occur as a strategy of self-protection[10,98]. While most frequently reported with parental neglect or maltreatment[99,100], our data suggest that similar processes may be at play in the face of maternal everyday stress. This capacity to adapt can be considered as resilience to adverse environments.

## Stress transmission between mothers and children in everyday life

Last, we aimed to examine factors that influence the degree to which prolonged perceptions of stress are shared between mothers and children. While children in our sample showed acute resonance with their mother's subjective and physiological stress[31], evidence of a covariance of prolonged stress within the same dyads could not be found. Therefore, we could not replicate the moderating role of trait empathy and closeness found in our recent[31] and other studies[16,17,29]. Of note, the present study focused on everyday empathic stress transmission as a source of chronic childhood stress and consequent cognitive impairment. However, it became apparent that stress resonance alone cannot account for maternal influences on child cognition. Other theories involve broader stressors shared within families such as poverty and discrimination[101,102] or altered parenting behavior due to stress experience[89].

## Limitations

There are several limitations to this study. First, our sample cannot be considered representative, as only healthy, German, and predominantly highly educated mother-child dyads participated. Across the whole sample, self-reported prolonged stress of mothers and children was low. Especially given the quadratic relationship observed within this limited stress range, future research should explore the circumstances under which stress promotes or compromises cognitive performance. To this end, we suggest extending findings to more diverse samples, broader ranges of chronic stress, and additional attachment constellations like father-child dyads.

Second, the questionnaire administered to assess children's perceived stress (PSS-C[71]) turned out to be of poor internal consistency. Although the PSS-C promised high comparability with the PSS and was considered an adequate measure of accumulated everyday stress in the investigated age group, its lack of reliability and validity compromised the quality of the results. Future research may further benefit from including physiological measures of long-term stress in addition to self-report scales.

Third, children were separated from their mothers upon arrival, despite physical proximity being a moderator of stress contagion[25,29]. While the separation was realized to prevent active emotional co-regulation, it is likely to have reduced empathic stress occurrence. As the hypothesized effects on child cognition are of indirect nature, future studies should maximize the likelihood of empathic stress transmission and recruit larger samples to increase sensitivity for smaller effects.

Last, the stressful nature of the unfamiliar environment and the cognitive tasks themselves may have been more salient for children than observing their mothers undergo the TSST. Although efforts were made to mitigate potential confounding effects, these factors cannot be entirely ruled out.

## Conclusions

Close relationships can be a double-edged sword. On the one hand, they provide us with support and protection in the face of stress. On the other hand, witnessing the stress of those close to us can inflict strain on ourselves.

The current results show that even moderately high acute and past-month stress in mothers can relate to impaired accuracy of their children in a cognitive flexibility task. Although results for working memory remained inconclusive and no evidence could be provided for the hypothesized connection between maternal and children's stress reactivity via empathic resonance, the current results make a viable contribution to the field of stress research in the family environment. Showing that very low or above average levels of maternal subjective and physiological stress are associated with impaired cognitive performance in children highlights the importance of the family environment for children's everyday outcomes. While the stress response can be adaptive, prolonged or severe exposure, even of secondary nature, bears the danger of long-term detrimental effects on body and brain[1]. Especially for children, in whom stress and its effects can have long-lasting developmental consequences[3,103], it is crucial to identify and tackle potential risks to strengthen resilience at an early age. Understanding the consequences of parental stress for children can inform future studies developing targeted strategies to help parents promote positive outcomes for their children.

## Data availability
The data used in this study, including cognitive performance metrics, physiological measures, and questionnaire responses, are publicly accessible at https://osf.io/kyrbh/.

## Code availability
Analysis scripts are publicly accessible at https://osf.io/kyrbh/.

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

## Acknowledgements

This study was funded by a project grant to Veronika Engert (EN 859/3-1) and Philipp Kanske (KA 4412/5-1) from the German Research Foundation (Deutsche Forschungsgemeinschaft). The funders had no role in study design, data collection and analysis, decision to publish or preparation of the manuscript. We thank Klara Steinberg for her valuable input and the Social Stress and Family Health Research Group at the Max Planck Institute for Human Cognitive and Brain Sciences, particularly Henrik Grunert, Elisabeth Murzik and Silvia Tydecks, for their support with running the study.

## Author contributions

V.E. and P.K. secured the funding and designed the experiment. V.E. planned the study and was responsible for study execution. J.U.B. supported data preparation. J.U.B. and J.R. supported data analysis. E.L. and I.G.L. analyzed the data and drafted the manuscript. V.E. supported the drafting of the manuscript. All authors critically revised the manuscript.

## Funding

## Competing interests

The authors declare no competing interests.
