## [Peer Review file · Communications Psychology]

Witnessing Their Mother's Acute and Prolonged Stress Affects Executive Functioning in Children

Corresponding Author: Ms Eileen Lashani

Version 0:

Decision Letter:

Dear Ms Lashani,

Thank you for your patience during the peer-review process. Your manuscript titled "From Support to Strain: Witnessing Their Mother's Acute and Chronic Stress Affects Executive Functioning in Children" has now been seen by 3 reviewers, and I include their comments at the end of this message. They find your work of interest but raised some important points. We are interested in the possibility of publishing your study in Communications Psychology, but would like to consider your responses to these concerns and assess a revised manuscript before we make a final decision on publication.

We therefore invite you to revise and resubmit your manuscript, along with a point-by-point response to the reviewers. Please highlight all changes in the manuscript text file.

Editorially, we consider it important that the revised manuscript address the analytic concerns raised by Reviewer 2 and 3. Please ensure the Introduction includes an appropriate literature review and conceptual framework to support the hypotheses. We appreciate that stress terminology has not always been applied consistently across disciplines, therefore we ask that you define the use of stress-related terminology upon first use of each term in your manuscript.

A number of the reported results do not reach conventional levels of statistical significance, without any evidence that you a priori set your alpha level to a value other than .05. Please remove claims of significance for any results with p values higher than .05. These results must be reported as non-significant nor may they be interpreted in the Discussion.

Please ensure you follow our statistical guidelines when reporting statistics (<https://www.nature.com/commspsychol/submit/submission-guidelines#statistical-guidelines>). Please note in particular our requirements for the reporting and interpretation of null-results. Non-significant findings derived from null-hypotheses significance tests should be reported in full, but may not be interpreted. Where you interpret null results, this interpretation must be based on Bayes Factors or equivalence tests.

I am attaching an Editorial Requests Table that details critical reporting requirements for the revised manuscript. Please attend to each item and ensure your manuscript is fully compliant. We are requesting that your manuscript aligns with these requirements as this facilitates the evaluation of your manuscript, reducing delays in re-review and potential future acceptance. If your revised manuscript is not aligned with these requests on major issues, such as those concerning statistics, it may be returned to you for further revisions without re-review. Additional information can be found in our style and formatting guide <https://www.nature.com/documents/commspsychol-style-formatting-guide-accept.pdf> Communications Psychology formatting guide.

Please use the following link to submit your
- revised manuscript,
- point-by-point response to the referees' comments,

- cover letter (as a separate document),
- the Editorial Policy Checklist (see below),
- the Reporting Summary (see below), and
- the completed Editorial Request Table (attached):

Link Redacted

Best regards,

Jennifer Bellingtier

Jennifer Bellingtier, PhD
Senior Editor
Communications Psychology

REVIEWER EXPERTISE:

Reviewer #1 Stress response, parent-child relationships, executive function

Reviewer #2 Stress response, parent-child relationships, executive function

Reviewer #3 Stress response, parent-child relationships, executive function

REVIEWER REPORTS:

Reviewer #1 (Remarks to the Author):

Summary : In this study, the authors assessed whether 'mothers' psychological and biological markers of stress/stress exposure can act as a source of childhood stress' and impair cognitive function. The authors assigned mothers to either a stress or control condition and assessed cognitive flexibility and working memory in their children after they had observed their mothers being exposed to the laboratory stress or control condition. Subjective stress, heart rate and salivary cortisol were measured in both mother and child. The results showed that children's cortisol levels increased while observing their mothers being exposed to the laboratory stressor and this cortisol increase was associated with impairment in child's cognitive flexibility. Cognitive impairment in child followed a quadratic relationship with higher child's error rate at very low and high acute maternal stress and changes as a function of acute and chronic maternal stress. The authors interpret these results as showing that both acute and chronic maternal stress have a significant impact on their child's cognitive processing.

Review : This is an interesting paper that taps on an important topic, ie. potential spillover effects of maternal stress on children and its impact on the development of child's cognitive processing. The authors recruited a relatively large sample size and assessed various psychological and biological markers of stress in children and their mothers. However, there are some points in the literature review, methodology and interpretation of results that might benefit from major revisions from the authors.

Introduction :

-I found the introduction to be a bit disorganized, with paragraphs transitioning from notions of resilience, to acute stress, to chronic stress, back to resilience, etc. The introduction would need a complete reorganization in order to offer a better description of the exact goal of the study.

-Many appropriate references are missing when describing important psychoneuroendocrine notions. For example, on line 74, the authors report that 'The sharing of stress in mother-child dyads has been demonstrated in two naturalistic studies; one with six-year-olds, the other with adolescents (Pratt et al., 2017; Papp et al, 2009). However, there are many other studies published much earlier than 2009 that reported stress resonance effects in mother-child dyads (although the term 'resonance' was not used at the time and authors were referring to 'spillover effects'). The authors must refer to the appropriate authors when describing important notions and not just the most recent papers on the subject. As well, by stating that there are only two studies that assessed resonance effects in mother-child dyads is inappropriate so care must be taken

to appropriately refer to the scientific literature on this important topic. The same comment applies to the information provided on line 88 to 94. There are many other studies than the ones cited that assessed how children resonate with their stressed mothers in terms of cortisol.

-Throughout the paper, the authors adopt a 'causal mechanism approach of measures' that is inappropriate. One example is the following sentence on lines 119 and 120. It reads like this : ' The present study investigated both cognitive flexibility and working memory of 8- to 12-year-old children after exposure to their mothers' stress'. In the protocol used by the authors, the children were not exposed to their 'mother's stress'. They were observing their mothers being exposed to a laboratory stressor. The nuance is extremely important. It is not because one is exposed to a laboratory stressor that one is necessarily stressed. There exists tremendous variability in cortisol response to laboratory stressors and the TSST is certainly no an exception to this. Consequently, the authors make the frequent mistake of associating 'being exposed to a laboratory stressor' to 'being stressed'. Yet, most studies show that this is not the case. Another mistake frequently made is to state that because someone presents a significant increase in cortisol levels, this person is stressed. Again, this is not the case as more and more studies are showing a lack of correlation between subjective and biological markers of stress. I am not raising these points to say that the protocol used by the authors is inappropriate. The protocol is good, but the problem lies in the interpretation of the measured used in the study. Consequently, I would suggest a complete reorganization of the paper in order to better reflect the measured used without over-interpreting them. For example, the sentence on line 119-120 would be a better representation of the measured by stating that : ' The present study investigated both cognitive flexibility and working memory of 8- to 12-year-old children after they observed their mothers being exposed to a laboratory stressor'.

-Finally in the introduction, the link between acute and chronic stress in mothers and their impact on children is very weak. There are huge differences between the neuroendocrine processes that underlie acute and chronic stress and it is not clear at all how they correlate. Moreover, many studies show that cortisol response to acute stress in humans is not a marker of chronic stress and that different mechanisms are at play while determining their impact on behavior and/or cognitive function. Consequently, I do not see how Hypothesis #2 described on lines 152 and 153 ('Second, we predicted that chronic stress of children and mothers would influence the effect of the acute experimental condition, differentially affecting children's cognitive performance') is viable in endocrinological terms. A much better rationale for assessing acute and chronic stress should be provided in the introduction of the paper.

Method

-The authors assessed chronic stress in mothers (and children) using the Perceived Stress Scale. However, this scale is a measure of perceived recent (within a month in general) stress and not chronic stress. Consequently, the measure of 'chronic' stress in the present study (and the interpretation of the results on this measure) is thus not in line with research on chronic stress in humans.

-The authors deposited the analysis scripts on OSF but unfortunately, they did not preregister their study protocol and analysis plan. This is unfortunate as we can take for granted that the authors were aware about the importance of open science practices when developing their study (given that they deposited the analysis script on OSF) and yet, they chose not to preregister their study protocol and analysis plan. Such a preregistration would have ensured readership that all hypotheses and analyses performed in the paper were developed before the statistical analyses were performed and that results were analyzed according to the registered plan. The authors might wish to provide a rationale for their choice not to preregister the study in the new version of their manuscript.

-The section on statistical analysis needs work as the authors do not differentiate between mother and child stress measures. Consequently, results are difficult to understand. For example on lines 247 and 248, the authors state that 'Linear mixed models were employed to examine the relationships between stress measures and cognitive performance'. Here, the authors refer to the stress measures (which is an inappropriate term, see my previous comment) of the mother or the child?'. The section contains a lot of these types of sentences and should be reorganized to appropriately refer to the mother or the child's measures.

-In the section on analysis of results, the authors refer to the 'children's stress reactivity'. Yet, the children were not exposed to the laboratory stressor. The authors measured cortisol in children who observed their mothers being exposed to the laboratory stressor. The difference here is huge and it is not because a child observe his/her mother being exposed to a laboratory stressor that the child is necessarily stressed.

-The authors performed a very large numbers of analyses and yet, they consider that the analyses did not need correction (line 273). I tend not to agree with this and the authors might wish to provide a very good rationale for this choice.

Discussion

-Again, there are many instances of overinterpretation of measures ('mother's stress', 'children stress', etc) in the discussion of the paper and this has a significant impact on the interpretation of the results. The discussion should be rewritten with careful consideration of the exact measured used while avoiding overinterpretation of the meaning of each of these measures.

-Another example of overinterpretation refers to the notion of resilience. In the introduction of the paper as well as in the discussion, the authors state that the results they obtain contribute to the understanding of children's stress resilience and yet, the authors did not measure resilience in this population.

-In the discussion of the results obtained, the authors ask the important question that emerges from their study, ie. How can maternal stress impact child cognition if there is no acute resonance between mother and child as reported in the results section? Here, the authors propose that the answer lies in chronic stress. However, as I stated previously, the Perceived Stress Scales does not measure chronic stress but rather, perception of recent stress. It will thus be important for the authors to offer a clear rationale of the obtained results in line with the appropriate constructs measured by the questionnaires they used in the present study.

Reviewer #2 (Remarks to the Author):

In this experimental study, the authors examined effects of observing a mother's acute stress via the Trier Social Stress Test on cognitive flexibility and working memory in children age 8-12. The role of mother's chronic stress and children's own physiological stress response were also measured. Results showed that children's cognitive flexibility was impaired when their own chronic stress was high, but the mother's acute and chronic stress had greater impacts on children's cognition. These associations were quadratic, not linear, such that medium levels of mother stress was linked to better performance in their children. Overall, this is an innovative and interesting study that adds to our knowledge of the impacts of an attachment figure's acute and chronic stress on children. Data analysis scripts are published on OSF. I am impressed with the manuscript as a whole, and suggest a few revisions below.

1. My biggest "critique" is there is a lot going on in this manuscript, and at times it is hard to keep track of all the findings and what the authors consider to be most important. Some streamlining of the overall narrative would be helpful.
2. H2 does not specify direction of influence—is this hypothesis exploratory?
3. Were there group differences on PSS or any other trait measures? e.g. I could imagine mothers who had just undergone the stress condition might report higher chronic stress, which would muddy the results.
4. I would like to see more discussion of the result "Showing that very low or above average levels of maternal stress are associated with impaired cognitive performance in children." Maybe I missed it, but I wasn't left with a clear understanding of why the authors think this relationship was quadratic and not linear.

Reviewer #3 (Remarks to the Author):

This study examined the role of child stress and mother stress on EF in healthy children. The study is very strong in most ways and clearly of interest to the readership. The writing is great, the study design is clearly well thought and executed, and the analysis/interpretation are strong. The results are very interesting and a strong contribution to the literature. I only have minor comments regarding statistical interpretation/reporting.

- I strongly disagree with reporting results that are $p \geq .05$ as "marginally significant". The sample size is not particularly small, and any results $> .05$ should not be interpreted. This is particularly concerning as the authors interpret findings $> .05$ in select results sections. $.05$ is already too lenient
- Related, the results narrative needs to be better supported by specific statistical findings. the narrative right now largely makes a statement with a vague reference to a figure or table, without specific description of where to find the statistic. I am particularly concerned about the reporting of p-values in the results section as the authors have interpreted findings $p > .05$ but do not specify in the narrative.
- Are there multicollinearity issues with so many variables in each regression model? How was this addressed if variables strongly correlated to one another were loaded into the same model?
- Figures do not list any relevant statistics or specify what task is referred to in the error rate.

EDITORIAL POLICIES

We ask that you ensure your manuscript complies with our editorial policies and reporting requirements.

To that end, we require revised manuscripts to be accompanied by two completed items: a reporting summary that collects information on study design and procedure, and an editorial policy checklist that verifies compliance with all required editorial policies.

- <https://www.nature.com/documents/nr-reporting-summary.zip>>Nature Research Reporting Summary
- <https://www.nature.com/documents/nr-editorial-policy-checklist.pdf>>Editorial Policy Checklist

All points on the policy checklist must be addressed. Your revised manuscript can only be sent back to the referees if these

checklists are completed and uploaded with the revision.

Notes: If you have submitted a Stage 1 Registered Report, Review, Primer, Comment, or Perspective you do not need to submit these forms. If you have already submitted these forms, you may disregard this request.

Version 1:

Decision Letter:

Dear Ms Lashani,

Your manuscript titled "From Support to Strain: Witnessing Their Mother's Acute and Prolonged Stress Affects Executive Functioning in Children" has now been seen by our reviewers, whose comments appear below. In light of their advice I am delighted to say that we are happy, in principle, to publish a suitably revised version in Communications Psychology.

We therefore invite you to revise your paper one last time to address the remaining concerns of our reviewers and a list of editorial requests. At the same time we ask that you edit your manuscript to comply with our format requirements and to maximise the accessibility and therefore the impact of your work.

EDITORIAL REQUESTS:

SUBMISSION INFORMATION:

OPEN ACCESS:

At acceptance, you will be provided with instructions for completing the open access licence agreement on behalf of all authors. This grants us the necessary permissions to publish your paper. Additionally, you will be asked to declare that all required third party permissions have been obtained, and to provide billing information in order to pay the article-processing

charge (APC).

* **DATA AVAILABILITY:**

Link Redacted

Best regards,

Jennifer Bellingtier

Jennifer Bellingtier, PhD
Senior Editor
Communications Psychology

REVIEWERS' EXPERTISE:

Reviewer #2 Stress response, parent-child relationships, executive function
Reviewer #3 Stress response, parent-child relationships, executive function

REVIEWERS' COMMENTS:

Reviewer #2 (Remarks to the Author):

The authors have made extensive changes to the manuscript to clarify the rationale for the study and further specify their hypotheses and analyses. The manuscript is much stronger than it was previously. All my concerns have been addressed.

Reviewer #3 (Remarks to the Author):

The authors have appropriately addressed my concerns.

From Support to Strain: Witnessing Their Mother's Acute and Prolonged Stress Affects Executive Functioning in Children

Dear Editor and Reviewers,

We are grateful for the chance to revise our manuscript. What follows is a detailed letter that describes how we handled all suggestions made. We believe these changes have much strengthened our manuscript and are very thankful for the guidance. All changes are marked in yellow in the revised manuscript version.

Reviewer #1

Summary: In this study, the authors assessed whether 'mothers' psychological and biological markers of stress/stress exposure can act as a source of childhood stress' and impair cognitive function. The authors assigned mothers to either a stress or control condition and assessed cognitive flexibility and working memory in their children after they had observed their mothers being exposed to the laboratory stress or control condition. Subjective stress, heart rate and salivary cortisol were measured in both mother and child. The results showed that children's cortisol levels increased while observing their mothers being exposed to the laboratory stressor and this cortisol increase was associated with impairment in child's cognitive flexibility. Cognitive impairment in child followed a quadratic relationship with higher child's error rate at very low and high acute maternal stress and changes as a function of acute and chronic maternal stress. The authors interpret these results as showing that both acute and chronic maternal stress have a significant impact on their child's cognitive processing.

Review: This is an interesting paper that taps on an important topic, ie. potential spillover effects of maternal stress on children and its impact on the development of child's cognitive processing. The authors recruited a relatively large sample size and assessed various psychological and biological markers of stress in children and their mothers. However, there are some points in the literature review, methodology and interpretation of results that might benefit from major revisions from the authors.

We wish to thank the reviewer for this overall positive assessment of our paper. Please find our detailed answers to the issues raised below.

Introduction

- I found the introduction to be a bit disorganized, with paragraphs transitioning from notions of resilience, to acute stress, to chronic stress, back to resilience, etc. The introduction would need a complete reorganization in order to offer a better description of the exact goal of the study.
 - **Thank you for raising this important issue. We agree that the introduction initially presented a large number of interrelated constructs, which may have led to some disorganization. To address this, we have restructured the introduction as follows:**

1. **We now begin by defining the key concepts of stress and resilience (lines 56 to 62), which form the foundation of our study. This sets a clear context for the subsequent elaborations.**
 2. **We then provide an overview of the broader study concept, outlining how these constructs are interrelated (lines 63 to 72).**
 3. **Next, we transition to a general review of empathic stress transmission, then focus specifically on stress transmission within the family, gradually narrowing down to the transmission of stress from mothers to their children (lines 71 to 107).**
 4. **With the context of stress transmission established, we introduce the specific dyadic empathic stress paradigm used in our study, explaining its relevance to our research questions (lines 108 to 123).**
 5. **Following this, we delve into the effects of acute and chronic stress on cognitive functioning, particularly in children, to provide the necessary background for our hypotheses (lines 124 to 156).**
 6. **Finally, we conclude the introduction by summarizing the key points, pulling together the threads of empathic stress transmission and its impact on cognition, leading up to our hypotheses (lines 157 to 178).**
 - **We hope that this reorganization provides a clearer and more logical flow of the introduction, making the rationale for our hypotheses more compelling.**
- Many appropriate references are missing when describing important psychoneuroendocrine notions. For example, on line 74, the authors report that 'The sharing of stress in mother-child dyads has been demonstrated in two naturalistic studies; one with six-year-olds, the other with adolescents (Pratt et al., 2017; Papp et al, 2009). However, there are many other studies published much earlier than 2009 that reported stress resonance effects in mother-child dyads (although the term 'resonance' was not used at the time and authors were referring to 'spillover effects'). The authors must refer to the appropriate authors when describing important notions and not just the most recent papers on the subject. As well, by stating that there are only two studies that assessed resonance effects in mother-child dyads is inappropriate so care must be taken to appropriately refer to the scientific literature on this important topic. The same comment applies to the information provided on line 88 to 94. There are many other studies than the ones cited that assessed how children resonate with their stressed mothers in terms of cortisol.
 - **To provide a more comprehensive review of the existing literature regarding shared stress between parents and their children, we extended our references to earlier research and studies from broader contexts. In the lines 87 to 93, we included: Boss (1987) and Larson & Almeida (1999) for stress transmission within families; Cummings & Davies (2002) and Matjasko & Feldman (2005) for spillover effects from parents on their children; Feldman (2007), Field (1981), and Sethre-Hofstadt et al. (2002) for physiological covariation between parents and children during stressful situations. Additionally, we ensured not to make inappropriate claims and changed our wording accordingly, e.g. by referring to “meaningful examples” instead of “meaningful exceptions” (lines 104 to 105). Should you be under the impression that we missed a specific key citation, we would appreciate your suggestions and would be happy to include it.**

- Throughout the paper, the authors adopt a 'causal mechanism approach of measures' that is inappropriate. One example is the following sentence on lines 119 and 120. It reads like this : ' The present study investigated both cognitive flexibility and working memory of 8- to 12-year-old children after exposure to their mothers' stress'. In the protocol used by the authors, the children were not exposed to their 'mother's stress'. They were observing their mothers being exposed to a laboratory stressor. The nuance is extremely important. It is not because one is exposed to a laboratory stressor that one is necessarily stressed. There exists tremendous variability in cortisol response to laboratory stressors and the TSST is certainly no an exception to this. Consequently, the authors make the frequent mistake of associating 'being exposed to a laboratory stressor' to 'being stressed'. Yet, most studies show that this is not the case. Another mistake frequently made is to state that because someone presents a significant increase in cortisol levels, this person is stressed. Again, this is not the case as more and more studies are showing a lack of correlation between subjective and biological markers of stress. I am not raising these points to say that the protocol used by the authors is inappropriate. The protocol is good, but the problem lies in the interpretation of the measured used in the study. Consequently, I would suggest a complete reorganization of the paper in order to better reflect the measured used without over-interpreting them. For example, the sentence on line 119-120 would be a better representation of the measured by stating that : ' The present study investigated both cognitive flexibility and working memory of 8- to 12-year-old children after they observed their mothers being exposed to a laboratory stressor'.

 - We absolutely agree with the reviewer that the manuscript would benefit from using more precise terminology regarding our key measures of stress. We addressed this comment in two ways. First, we now define what we refer to when talking about “stress” at the very beginning of the introduction, distinguishing between the subjective experience of stress and the physiological stress response (lines 59 to 62). While the subjective and physiological stress responses do not necessarily correlate, both are considered measures of “stress”, as backed up by the appropriate literature (e.g., Cohen et al., 1997). Second, we revised our wording throughout the manuscript to avoid confounding the stress we measured and potential stress we may have raised due to our methodological setup (e.g., changing “after exposure to their mothers’ stress” to “after observing their mothers being exposed to either a laboratory stressor or a stress-free control”, lines 138 to 140). We hope that these adjustments improve the accuracy of our report.**
- Finally in the introduction, the link between acute and chronic stress in mothers and their impact on children is very weak. There are huge differences between the neuroendocrine processes that underlie acute and chronic stress and it is not clear at all how they correlate. Moreover, many studies show that cortisol response to acute stress in humans is not a marker of chronic stress and that different mechanisms are at play while determining their impact on behavior and/or cognitive function. Consequently, I do not see how Hypothesis #2 described on lines 152 and 153 ('Second, we predicted that chronic stress of children and mothers would influence the effect of the acute experimental condition, differentially affecting children's cognitive performance') is viable in endocrinological terms. A much better rationale for

assessing acute and chronic stress should be provided in the introduction of the paper.

- **We thank the reviewer for addressing this issue. To provide a more thorough rationale for the inclusion of a marker of prolonged stress perceptions, we elaborated on the link between acute and chronic stress (lines 118 to 123). We hope that these additions, which draw from established literature in the field, sufficiently explain the consideration of the chosen psychophysiological stress markers (not limited to cortisol, where we acknowledge that the literature to date is inconclusive regarding the exact patterns following chronic or traumatic stress).**

Method

- The authors assessed chronic stress in mothers (and children) using the Perceived Stress Scale. However, this scale is a measure of perceived recent (within a month in general) stress and not chronic stress. Consequently, the measure of 'chronic' stress in the present study (and the interpretation of the results on this measure) is thus not in line with research on chronic stress in humans.
 - **We thank the reviewer for pointing this out. We have adjusted our wording and interpretation of results to reflect our measurement with the PSS more accurately. Statements that indicated that we measured “chronic” stress have been replaced by “stress in the past month” or, to enhance readability and simplicity, “prolonged”.**
- The authors deposited the analysis scripts on OSF but unfortunately, they did not preregister their study protocol and analysis plan. This is unfortunate as we can take for granted that the authors were aware about the importance of open science practices when developing their study (given that they deposited the analysis script on OSF) and yet, they chose not to preregister their study protocol and analysis plan. Such a preregistration would have ensured readership that all hypotheses and analyses performed in the paper were developed before the statistical analyses were performed and that results were analyzed according to the preregistered plan. The authors might wish to provide a rationale for their choice not to preregister the study in the new version of their manuscript.
 - **We are aware that you are addressing an important point regarding the preregistration. We agree that preregistration would have improved the transparency of our analyses. The current manuscript is based on a secondary data analysis conducted within the context of a Master's thesis (the primary study was preregistered at <https://osf.io/f3wk> and published by Blasberg et al. in 2023). Unfortunately, when the Master's thesis focusing on cognitive data was initially proposed and planned, we did not conduct a preregistration. To nevertheless provide as much transparency as possible and facilitate replications of our findings, all scripts have been deposited on OSF and we are willing to share data with other researchers upon reasonable request.**
- The section on statistical analysis needs work as the authors do not differentiate between mother and child stress measures. Consequently, results are difficult to understand. For example on lines 247 and 248, the authors state that 'Linear mixed models were employed to examine the relationships between stress measures and cognitive performance'. Here, the authors refer to the stress measures (which is an

inappropriate term, see my previous comment) of the mother or the child?'. The section contains a lot of these types of sentences and should be reorganized to appropriately refer to the mother or the child's measures.

- **This is indeed an important issue. We revised ambiguous formulations and now explicitly specify whether a variable refers to children, mothers, or both throughout the manuscript.**
- In the section on analysis of results, the authors refer to the 'children's stress reactivity'. Yet, the children were not exposed to the laboratory stressor. The authors measured cortisol in children who observed their mothers being exposed to the laboratory stressor. The difference here is huge and it is not because a child observe his/her mother being exposed to a laboratory stressor that the child is necessarily stressed.
 - **We hope that our interpretations of the “stress response” are more coherent already after initially defining our conceptualization of stress in the introduction. Additionally, we have revised the wording with respect to the actual child experience (i.e., observing their mothers undergo a stressful experience) throughout the manuscript. In this context, we would also like to highlight the conceptualization of a physiologically significant cortisol release in response to the experimental condition, which we defined under *Group Differences* (lines 269 to 272) and which follows the established norms (i.e., an increase in cortisol levels of at least 1.5 nmol/l from baseline to peak concentrations; Miller et al.,2013). The manipulation check described there further supports our argument that observing their mothers undergoing stress can act as a stressor for children because it leads to the release of cortisol above the 1.5 nmol/l threshold. For this reason, based on an empathic stress framework, we investigate and report children’s psychophysiological “stress response” to observing their mothers in a stressful situation.**
- The authors performed a very large numbers of analyses and yet, they consider that the analyses did not need correction (line 273). I tend not to agree with this and the authors might wish to provide a very good rationale for this choice.
 - **We understand that whether or not to correct for multiple comparisons is always a debated issue. Based on the number of conducted analyses, we thoroughly considered whether correction for multiple comparisons was necessary. We followed the recommendations described in a recent article by García-Pérez (2023) and provided an explanation of the underlying rationale in the highlighted paragraph (lines 318 to 324). Specifically, García-Pérez argues that when analyses do not test the same "omnibus null hypothesis," the application of multi-comparison correction may not be necessary. As you have also pointed out in your previous comments, the measures of subjective and physiological stress reactivity, as well as acute and chronic stress, do not necessarily reflect identical underlying processes, and often fail to show associations. Additionally, performance outcomes such as speed and accuracy may follow distinct patterns, and cognitive functions like working memory and cognitive flexibility exhibit differential associations and vulnerabilities to psychophysiological stress, as discussed in the introduction. Because these analyses examine distinct aspects of a broader hypothesis, rather than a single overarching one, we believe**

that correcting for multiple comparisons would be overly conservative and potentially obscure meaningful findings. To address the concern of being too liberal in our interpretations, we have taken a more stringent approach by removing all interpretations related to marginally significant findings, considering them as “non-significant.” We hope that this balanced approach, along with the rationale provided, addresses the conservative-liberal trade-off appropriately.

Discussion

- Again, there are many instances of overinterpretation of measures ('mother's stress', 'children stress', etc) in the discussion of the paper and this has a significant impact on the interpretation of the results. The discussion should be rewritten with careful consideration of the exact measures used while avoiding overinterpretation of the meaning of each of these measures.
 - **Hoping to have fixed the issue of overinterpretation, we revised our wording and specified interpretations where needed, referring to physiological or subjective stress reactivity, respectively. This way, our interpretations should be coherent within the manuscript and in line with other research on the topic.**
- Another example of overinterpretation refers to the notion of resilience. In the introduction of the paper as well as in the discussion, the authors state that the results they obtain contribute to the understanding of children's stress resilience and yet, the authors did not measure resilience in this population.
 - **Thank you for pointing this out. It is correct that we did not explicitly measure resilience in form of a trait (e.g., with a questionnaire). As highlighted in the introduction (lines 56 to 59), we instead conceptualized resilience as a measurable outcome following a challenging or adverse situation, which is a common approach in this field (Masten et al., 1990, Masten & Obradović, 2007). That is, we probe resilience by assessing children's cognitive performance (outcome) following the observation of their mothers undergoing a stressful task or reporting prolonged stress (challenge/ adversity). To emphasize this approach, we changed our wording following the resilience and stress definitions in the introduction and added the resilience perspective to the opening sentence of the discussion (lines 449 to 452).**
- In the discussion of the results obtained, the authors ask the important question that emerges from their study, ie. How can maternal stress impact child cognition if there is no acute resonance between mother and child as reported in the results section? Here, the authors propose that the answer lies in chronic stress. However, as I stated previously, the Perceived Stress Scales does not measure chronic stress but rather, perception of recent stress. It will thus be important for the authors to offer a clear rationale of the obtained results in line with the appropriate constructs measured by the questionnaires they used in the present study.
 - **Absolutely! Next to adjusting our wording regarding the influence of the PSS(-C) measure throughout the manuscript (into prolonged rather than chronic stress), we addressed this point by providing a potential explanation for the effect of stress perceptions on the acute stress response. We did so by linking our argument for the influence of long-**

term stress to general psychophysiological reactivity patterns depending on the appraisal of a situation (lines 533 to 539). Further, we acknowledge in our limitations section that it would be important for future studies to include a reliable, ideally also physiological, measure of long-term stress (lines 592 to 593).

Reviewer #2

In this experimental study, the authors examined effects of observing a mother's acute stress via the Trier Social Stress Test on cognitive flexibility and working memory in children age 8-12. The role of mother's chronic stress and children's own physiological stress response were also measured. Results showed that children's cognitive flexibility was impaired when their own chronic stress was high, but the mother's acute and chronic stress had greater impacts on children's cognition. These associations were quadratic, not linear, such that medium levels of mother stress was linked to better performance in their children. Overall, this is an innovative and interesting study that adds to our knowledge of the impacts of an attachment figure's acute and chronic stress on children. Data analysis scripts are published on OSF. I am impressed with the manuscript as a whole, and suggest a few revisions below.

We very much thank the reviewer for this positive assessment of our study.

1. My biggest "critique" is there is a lot going on in this manuscript, and at times it is hard to keep track of all the findings and what the authors consider to be most important. Some streamlining of the overall narrative would be helpful.
 - **We absolutely agree, and the same point was also raised by another reviewer. We recognize that the manuscript covers a wide range of constructs and findings, which may have contributed to the complex narrative. To address this concern, we have streamlined the overall narrative, especially by restructuring the introduction. We now begin the introduction by defining the core concepts of stress and resilience to provide a clear foundation for the study (lines 56 to 62). Next, we organized the introduction to follow a more logical flow, starting with a broad overview of empathic stress transmission and gradually narrowing down to specific findings, particularly on stress transmission from mothers to children. Afterwards, we review findings on acute and chronic stress effects on cognition and tie together these two big blocks on empathic stress transmission and stress effects on cognition. We hope that these changes make the deduction of our hypotheses more straightforward.**
In the results section, we attempted to provide a helpful structure by using subheadings and providing a short summary at the end of each paragraph.
Likewise, the discussion is now structured by subheadings for each block of analyses, after a broader summary of the overall research questions and findings. In addition, we elaborated on the link between acute stress and perceived prolonged stress to improve the flow of our arguments (lines 533 to 539).
We hope these revisions make the manuscript more cohesive and easier to follow, allowing readers to grasp the most important aspects of our research.
2. H2 does not specify direction of influence—is this hypothesis exploratory?
 - **Thank you for spotting this vagueness in the phrasing of H2. The hypothesis is indeed exploratory, as the prolonged experience of stress of both mother and child can be assumed to either lead to sensitizing or attenuating alterations of their stress systems. To make this clearer, we**

added a paragraph to the introduction, which provides references for the different potential outcomes (lines 118 to 123). We further adjusted our phrasing of the hypothesis to make the exploratory nature more explicit (“Second, we explored whether prolonged experiences of everyday stress in children and mothers would influence the effect of the acute experimental condition, [...]”).

3. Were there group differences on PSS or any other trait measures? e.g. I could imagine mothers who had just undergone the stress condition might report higher chronic stress, which would muddy the results.
 - **You are raising an important point with this question. We have now reported an additional ANOVA assessing group differences in the PSS and PSS-C between stress and control groups (lines 282 to 283, and 343 to 347). As the groups did not statistically differ, we conclude that the stress manipulation did not significantly affect the report of prolonged stress. Furthermore, as detailed in Blasberg et al. (2023), other trait variables also showed no significant differences between the groups.**
4. I would like to see more discussion of the result "Showing that very low or above average levels of maternal stress are associated with impaired cognitive performance in children." Maybe I missed it, but I wasn't left with a clear understanding of why the authors think this relationship was quadratic and not linear.
 - **This is an absolutely valid point! In general, we would like to stay transparent in stating that our a priori predictions did not focus on the quadratic or linear nature of the relationship between stress markers and outcomes but rather took both into account without specific expectations. In lines 308 to 310, we provide a short explanation of why we exploratively included quadratic terms. For better visibility in the revision process, we have highlighted this sentence. When the analysis revealed that most stress markers indeed related to children's outcomes in a quadratic fashion, we discussed it accordingly in the section “Acute Stress Effects on Cognitive Performance”. For more clarity, we adjusted this section by more explicitly introducing the quadratic effects as an exploratory finding (“Through the exploratory second step of the data analysis, accounting for a possible quadratic relationship, [...]”).**

Reviewer #3

This study examined the role of child stress and mother stress on EF in healthy children. The study is very strong in most ways and clearly of interest to the readership. The writing is great, the study design is clearly well thought and executed, and the analysis/interpretation are strong. The results are very interesting and a strong contribution to the literature. I only have minor comments regarding statistical interpretation/reporting.

We very much appreciate your positive stance on our manuscript!

- I strongly disagree with reporting results that are $p \geq .05$ -.09 as “marginally significant”. The sample size is not particularly small, and any results $> .05$ should not be interpreted. This is particularly concerning as the authors interpret findings $> .05$ in select results sections. .05 is already too lenient
 - **We completely understand your concern regarding the interpretation of effects $p > .05$. Following your suggestion, we have therefore discarded all interpretations of marginal findings, now reporting them as non-significant, in line with more stringent statistical standards.**
- Related, the results narrative needs to be better supported by specific statistical findings. the narrative right now largely makes a statement with a vague reference to a figure or table, without specific description of where to find the statistic. I am particularly concerned about the reporting of p-values in the results section as the authors have interpreted findings $p > .05$ but do not specify in the narrative.
 - **Thank you for pointing this out. In response, we have addressed the issue of findings with $p > .05$, which were previously labeled as marginal or trend significant, by reclassifying them as non-significant. As a result, there are no remaining interpretations of effects that are not statistically significant in the results section. To comply with common reporting practices, avoid redundancy, and enhance readability, we have refrained from repeating the detailed statistics that are already presented in the tables within the main text. However, we recognize the importance of clarity in the narrative, and significant effects that are otherwise only reported in the supplementary materials are fully described in the main text, as highlighted. We have also ensured that the table structure is clear and that each paragraph in the results section now includes a corresponding reference to the relevant table. We hope that this approach provides a sufficient guide for readers to follow our narrative and locate the specific statistical details easily.**
- Are there multicollinearity issues with so many variables in each regression model? How was this addressed if variables strongly correlated to one another were loaded into the same model?
 - **To address multicollinearity, we z-standardized all continuous predictors. Additionally, we now double-checked Variance Inflation Factors (VIF) and found them to be < 5 for all variables in each model, indicating no problematic multicollinearity. We hope that the revised description of these steps in the methods section sufficiently addresses the raised concerns (lines 316 to 318).**

- Figures do not list any relevant statistics or specify what task is referred to in the error rate.
 - **We apologize for the unclarity. In fact, the error rate always refers to the category-switching task, as accuracy in the n-back task was indexed by d' as described in the Methods section (lines 279 to 281). To prevent any potential confusion, we have revised the figures to ensure that all figure notes now clearly specify the task associated with the results. Additionally, we have reviewed the figures to confirm they meet all relevant requirements. This includes displaying standard errors, all individual data points, and the number of participants.**

Dear Reviewers,

Thank you for your thoughtful and constructive feedback on our manuscript, "Witnessing Their Mother's Acute and Prolonged Stress Affects Executive Functioning in Children." We greatly appreciate your comments, which have helped us strengthen the paper. We are pleased that the changes satisfactorily addressed all your concerns.

We would like to express our sincere gratitude for your time and effort in reviewing our manuscript.

Sincerely,

Eileen Lashani (on behalf of all authors)

REVIEWERS' COMMENTS

Reviewer #2

The authors have made extensive changes to the manuscript to clarify the rationale for the study and further specify their hypotheses and analyses. The manuscript is much stronger than it was previously. All my concerns have been addressed.

Reviewer #3

The authors have appropriately addressed my concerns.